# Micelle Formation in Aqueous Solutions of the Cholesterol-Based Detergent Chobimalt Studied by Small-Angle Scattering

**DOI:** 10.3390/molecules28041811

**Published:** 2023-02-14

**Authors:** Oleksandr P. Artykulnyi, Katarina Siposova, Manfred Kriechbaum, Andrey Musatov, László Almásy, Viktor Petrenko

**Affiliations:** 1Faculty of Physics, Taras Shevchenko Kyiv National University, 01601 Kyiv, Ukraine; 2Department of Biophysics, Institute of Experimental Physics, Slovak Academy of Sciences, 04001 Kosice, Slovakia; 3Institute of Inorganic Chemistry, Graz University of Technology, 8010 Graz, Austria; 4Institute of Energy Security and Environmental Safety, Centre for Energy Research, 1121 Budapest, Hungary; 5BCMaterials, Basque Center for Materials, Applications and Nanostructures, 48940 Leioa, Spain; 6Ikerbasque, Basque Foundation for Science, 48009 Bilbao, Spain

**Keywords:** Chobimalt, cholesterol-based detergent, sugar-based detergent, amphiphilic molecule, micelle, small-angle neutron scattering, small-angle X-ray scattering

## Abstract

The structure and interaction parameters of the water-soluble cholesterol-based surfactant, Chobimalt, are investigated by small-angle neutron and X-ray scattering techniques. The obtained data are analyzed by a model-independent approach applying the inverse Fourier transformation procedure as well as considering a model fitting procedure, using a core-shell form factor and hard-sphere structure factor. The analysis reveals the formation of the polydisperse spherical or moderately elongated ellipsoidal shapes of the Chobimalt micelles with the hard sphere interaction in the studied concentration range 0.17–6.88 mM. The aggregation numbers are estimated from the micelle geometry observed by small-angle scattering and are found to be in the range of 200–300. The low pH of the solution does not have a noticeable effect on the structure of the Chobimalt micelles. The critical micelle concentrations of the synthetic surfactant Chobimalt in water and in H_2_O-HCl solutions were obtained according to fluorescence measurements as ~3 μM and ~2.5 μM, respectively. In-depth knowledge of the basic structural properties of the detergent micelles is necessary for further applications in bioscience and biotechnology.

## 1. Introduction

Micellar solutions of amphiphilic molecules (detergents or surfactants) are classic examples of systems with self-organization behavior which are still the subjects of intensive investigations in the fields of colloidal chemistry, physical chemistry, and molecular physics as well as application-related research.

Thus, it is well-known that hydrophobic interactions play a key role in determining the native structure of proteins, which consist of both polar and nonpolar amino acids. However, unlike water-soluble proteins, in which the hydrophobic domains are protected from the aqueous environment, within membrane proteins, the hydrophobic domains are surrounded by lipids and, therefore, these proteins are not soluble in an aqueous solution. Different membrane-mimetic systems have been developed for use in the field of membrane-protein research. To study hydrophobic membranes or integral proteins in vitro, a special group of amphiphilic compounds called detergents comes to the rescue. Detergents are amphiphilic compounds that contain both polar (head) and hydrophobic (tail) groups. In an aqueous solution, the polar groups form hydrogen bonds with water molecules, while the hydrophobic groups self-aggregate into micelles of various types that are soluble in water. Micelles can be spheres, ellipsoids, or cylinders of various sizes, determined by the detergent head group structure and alkyl chain length [1]. In addition, detergents are capable of forming non-micellar structures, such as nanodiscs [2] and bicelles [3]. Micelle formation occurs at a detergent concentration at or above the critical micelle concentration (CMC). Micelles are used for the solubilization of membrane proteins during purification, crystallization, and a further in vitro analysis.

The development of novel cholesterol-based detergents with an improved membrane-protein stabilization efficiency is an actual direction of the current research in this area [4,5,6,7]. It is important to note that some detergents can strongly affect the structure of the protein, while other detergents keep the native structure of the protein unaltered [8]. Therefore, knowledge of the detergent properties and, in particular, knowledge of the size and shape of micelles and the intermicellar interaction is necessary for the right choice of detergent and conditions for further work with proteins.

A novel water-soluble derivative of cholesterol, β-cholesteryl-(maltosyl-β-(1,6)-maltoside), named Chobimalt (Figure 1), was synthesized previously by Howell et al. [9] to simulate insoluble cholesterol and, therefore, to be beneficial for the experimental modeling of the biological activity of cholesterol. However, despite the high potential of use, very little attention has been paid to this detergent so far. We believe there are at least two reasons for this. Firstly, in the original paper by Howell et al. [9], it was demonstrated that Chobimalt itself is not a very effective detergent for membrane extraction. However, when used as a cosurfactant, Chobimalt could associate with other detergents to form mixed micelles with unique properties. For example, in combination with the nonionic detergent dodecyl maltoside, it confers the long-term thermal stability of the solubilized human kappa opioid receptor type 1 (hKOR1), which indicates the role of cholesterol in the stabilization of the native proteins. Understanding protein stability is crucial for optimizing the protein purification, storage, structural studies, industrial application, etc. In addition, Chobimalt has recently been successfully used to elucidate the effect of a cholesterol-based detergent on amyloidogenesis, and an intriguing dose-dependent effect of Chobimalt on the formation of an insulin amyloid formation has been described [10]. Knowledge of the mechanism of modulation of amyloid protein fibrillization by amphiphilic compounds, in particular cholesterol, is important for understanding molecular events inside cells that lead to many neurodegenerative diseases, such as Alzheimer’s or Parkinson’s disease. Thus, in protein research, Chobimalt, like other detergents, can be used to study not only the highly hydrophobic membrane proteins but also water-soluble proteins. Consequently, the formation of micelles during protein fibrillogenesis should be considered among the factors determining the effectiveness of Chobimalt molecules. However, there is a lack of detailed information about the structural organization of Chobimalt micelles, which, in our opinion, is the second reason for the low attention to Chobimalt. The structural organization of Chobimalt micelles should be investigated under various conditions, in particular at an acidic pH, because in the in vitro experiments the acidic pH promotes amyloid aggregation, and therefore extensive studies have been conducted at a pH as low as pH 2.0 [11,12,13,14]. So, such parameters as the detergent micellar size and packing, detergent–detergent interaction, and hydrophilic–lipophilic balance are crucially important for Chobimalt applications and development.

In the present study, small-angle scattering methods were applied to investigate the micelles structure formed by Chobimalt in aqueous solutions. The small-angle neutron and X-ray scattering (SANS and SAXS) techniques are well-established and widely used for the structural investigation of micellar solutions [15,16,17]. Information about the structure of micelles and the intermicellar interaction can be derived from the small-angle scattering data by applying model-independent approaches, such as the Fourier transformation or various approximations on scale–transformed data. A more versatile and more often used approach is the model fitting, either for single-component solutions [18,19,20,21] or multicomponent systems, such as surfactants with polymers [22], suspensions of nanoparticles [23,24], or mixtures of proteins with micelles [25], among others.

In the present work, SANS and SAXS were used to determine the parameters of the structure and the interactions of the Chobimalt micelles in ultra-pure water and water of pH 2 (adjusted with HCl), over a range of detergent concentrations. The CMC of Chobimalt was also determined under the same conditions using a fluorescent probe.

## 2. Results and Discussion

### 2.1. Critical Micelle Concentration

The critical micelle concentration of Chobimalt dissolved either in H_2_O or water of pH 2 (hereinafter referred as to H_2_O-HCl) was measured using an extrinsic fluorescent probe 8-anilino-1-napthalene sulfonic acid (ANS). The formation of micelles is accompanied by the change in the spectroscopic behavior of the extrinsic fluorescent molecules; in the case of the ANS, the fluorescence significantly increases even at a low, picomolar concentration of micelles [26]. To determine the CMC of Chobimalt, the measurements of the ANS fluorescence at a constant concentration as a function of an increased Chobimalt concentration were carried out. As shown in Figure 2, at a low concentration of Chobimalt, up to ~2 μM, the fluorescence of the ANS did not change. However, starting from 2 to 3 μM, a rapid increase in the ANS fluorescence was observed. The breakpoint separating the rapid increase in the ANS fluorescence and linear region marks the CMC, the value of which was found to be ~2.5 μM in H_2_O. It was also found that the CMC of Chobimalt was not significantly affected by the presence of salt (Appendix A, Appendix A) or a low pH and was in the range of 2.5–3 µM (Figure 2). This result corresponds well to the previously published value of 3–4 μM observed by Howell et al., who studied the formation of Chobimalt micelles using 250 nM and 10 μM concentrations of the ANS [9].

### 2.2. Small-Angle Scattering

The experimental SANS curves of the Chobimalt micelles in D_2_O-DCl are shown in Figure 3. The scattering curves display a typical pattern of compact, i.e., not strongly elongated, particles with no apparent repulsive interaction.

The SAXS experiments were performed at three concentrations of Chobimalt above the CMC, in H_2_O, and in H_2_O-HCl, applied previously in a study of the effect of Chobimalt on the amyloidogenesis of protein [10]. The scattering patterns display a characteristic difference from those of the SANS scattering curves, marked by the presence of a strong oscillation, with a minimum at about q = 0.6 nm^−1^. This difference is due to the strong contrast between the micelle shell and the core in the case of X-rays. The experimental SAXS curves of the Chobimalt micelles with fitting functions are shown in Figure 4.

Firstly, the small-angle neutron and X-ray scattering data were analyzed by a model-independent approach, applying the inverse Fourier transformation (IFT) procedure developed by Glatter [27] and Moore [28]. This method requires only minor additional information to fit the small-angle scattering data, that is, the maximal particle size, which is determined by comparing the resulting pair distance distribution functions calculated for various assumed maximum micelle sizes. An IFT analysis was performed using the SASView software [29]. The pair distance distribution (PDD) functions, p(r), of the Chobimalt micelles obtained from the SANS and SAXS data are shown in Figure 5. The close-to-symmetrical shape of the PDD functions obtained from the SANS data indicates spherical-like particles [30], and the slight asymmetry with an elongated tail toward large distances r points to moderately elongated or ellipsoidal particles, which can be the typical shape of micelles in solution. From the SAXS data, typical PDD functions characteristic to core-shell particles are obtained, characterized by a secondary maximum at small distances (~3 nm) [31]. These results indicate the existence of core-shell particles or micelles. Therefore, the next step was to fit the experimental data by certain particle shape models and derive the structural information about the Chobimalt micelles based on the obtained model parameters.

The IFT analysis further provides the values of the scattering invariants, such as the radius of gyration, R_g_, forward scattering intensity, I_0_, and maximum size of the particle, D_max_. These values are collected in Table 1.

Thus, according to the shape of the obtained PDD functions (Figure 5), the small-angle scattering data of the Chobimalt solutions were interpreted as a scattering of spherical or ellipsoidal micelles with a possible core-shell structure. A further analysis was performed, employing established mathematical models of the micelle structure and interactions. The scattering intensity as a function of the momentum transfer for the dispersion of monodisperse particles can be described by the following equation:(1)$$I\left(q\right)=nP\left(q\right)S\left(q\right),$$
where n—particle number concentration, $P\left(q\right)$—form factor, and $S\left(q\right)$—structure factor.

The value of the scattering length density (SLD or *ρ*) in a molecular liquid or solid is determined by the equation
(2)$$SLD=\frac{1}{v_{m}}\underset{i}{\sum }b_{c,i}$$
where $b_{c}$ is the bound coherent scattering length of the compound atom *i* of the molecule, and $v_{m}$ is the molecular volume. The hydrophobic part of Chobimalt, represented by the cholesterol molecule, has SLD values of about 0.22 × 10^10^ cm^−2^ and 10.0 × 10^10^ cm^−2^ for the thermal neutrons and X-ray Cu K_α_ radiation, respectively, whereas the SLD values of the maltose compound has 1.66 × 10^10^ cm^−2^ and 13.96 × 10^10^ cm^−2^ for the neutrons and Cu K_α_ radiation. Because the hydrophobic and hydrophilic parts of the Chobimalt molecule have slightly different SLD values, while the polar part of the surfactant molecule has a comparable size to its nonpolar part, the core-shell model should be used to describe the micelle form factor.

The scattering amplitude *F*(q) of the core-shell sphere particle is calculated in the following way:(3)$$F\left(q\right)=\frac{3}{V_{mic}}\left[V_{core}\left(\rho_{core}-\rho_{shell}\right)\frac{j_{1}\left(qr\right)}{qr}+V_{shell}\left(\rho_{shell}-\rho_{solvent}\right)\frac{j_{1}\left(qh\right)}{qh}\right],$$
where $V_{mic}$; $V_{core}$; $V_{shell}$ are the volume of the entire micelle and the volume of its core and shell, respectively, $j_{1}\left(x\right)$ is the first-order Bessel function, r is the radius of the micelle core, h is the thickness of the shell, and $\rho_{core};\;\rho_{shell};\;\rho_{solvent}$ are the core, shell, and solvent SLD, respectively. The structure factor *S*(q) is calculated using the hard-sphere interaction potential with a Percus–Yevick closure [32], for which the volume fraction of micelles φ was used.

The variation in the form factor should be taken into account when considering the scattering of the polydisperse systems:(4)$$P\left(q\right)=\overset{\infty }{\underset{0}{\int }}f\left(x;\bar{x},\sigma \right){\left|F\left(q,x\right)\right|}^{2}dx,$$
where $f\left(x;\bar{x},\sigma \right)$ is the size distribution function with the mean size of particles $\bar{x}$, and σ is the root-mean-square deviation parameter of the distribution. For micellar systems, it is convenient to use the two-parametric Schulz distribution function, which allows the calculation of integral (4) explicitly [33]:(5)$$f\left(x\right)={\left(z+1\right)}^{z+1}{\left(x/\bar{x}\right)}^{z}\frac{\exp\left(-\left(z+1\right)x/\bar{x}\right)}{\bar{x}\Gamma \left(z+1\right)},$$
where $\Gamma \left(x\right)$ is the gamma function, and z=1−σx¯2 x¯σ¯2 is the measure of the width of the distribution [1]. The description of the structure factor for polydisperse systems is more complicated; therefore, the approximate function for the hard-sphere interaction potential with an effective radius of interaction is used. In this way, the scattering intensity (1) is modified for taking into account the effect of polydispersity [26]:(6)$$I\left(q\right)=nP\left(q\right)\left[1+\beta \left(q\right)\left(S\left(q\right)-1\right)\right],$$
where $\beta \left(q\right)$ is a correction factor for polydisperse or anisotropic systems that suppress the oscillations of the true structure factor in the experimental scattering intensity. Equation (6) was used to fit the data by the non-linear least squares method [16] by the Levenberg–Marquardt minimization algorithm, using the free software SasView [29]. The results of the model fitting of the experimental SANS and SAXS data are collected in Table 2, Table 3 and Table 4.

Based on the geometrical parameters of the micelles, consisting of a hydrophobic core and a hydrated maltose shell, the average aggregation number $N_{agg}$ can be calculated, which is the key characteristic of micelles of a particular surfactant under the given external conditions of the system. In our case of a core-shell micelle formed from cholesterol–maltose surfactant molecules, the aggregation number can be calculated from the following equation, considering a dry cholesterol core and water-containing maltose shell:(7)$$\frac{4}{3}\pi {\left(R_{core}+h\right)}^{3}=N_{agg}\left[v_{ch}+2\cdot v_{malt}+\frac{\rho_{shell}-\rho_{malt}}{\rho_{D_{2}O}-\rho_{malt}}v_{D_{2}O}\right]$$
where $v_{ch}=627\;{Å}^{3}$, $v_{malt}=345\;{Å}^{3}$, $v_{D_{2}O}=30\;{Å}^{3}$—molecular volumes of cholesterol, maltose, and water molecules, respectively [34,35]. The calculated values of the $N_{agg}$ of the Chobimalt micelles obtained from the SANS data are presented in Table 2, together with the characteristic parameters of the Chobimalt micelles in D_2_O and D_2_O-DCl for the polydisperse spherical micelle model.

As can be seen from Table 2, the parameter of the average micelle aggregation number increases monotonously with an increasing concentration and is slightly larger than the value $N_{agg}=$ 203 ± 29 obtained using a different method by Howell et al. [9]. The increased value of the fitted parameter SLD of the micelle shell (Table 2) compared to the calculated value for maltose SLD = 1.66 10^10^ cm^−2^ indicates hydration of the hydrophilic shell, which reaches about (ρshell−ρmalt)ρD2O−ρmalt ≈ 10% of the shell volume.

Alternatively to polydisperse spherical micelles, we also considered the model of ellipsoids of revolution with a hard-sphere interaction, without taking into account the possible polydispersity. This choice is justified because of the very similar effect of the polydispersity and a moderately elongated shape of the ellipsoidal micelles on the resulting small-angle scattering curves. The scattering form factor $P\left(q\right)$ of core-shell ellipsoid particles is calculated in the following way [16]:(8)$$P\left(q\right)=\frac{1}{V_{mic}}\;\overset{\pi /2}{\underset{0}{\int }}{\left[\frac{3V_{core}\left(\rho_{core}-\rho_{shell}\right)j_{1}\left(qR\left(R_{e},R_{p},\alpha \right)\right)}{qR\left(\alpha \right)}+\frac{3V_{shell}\left(\rho_{shell}-\rho_{solvent}\right)j_{1}\left(qR\left(R_{e}+h,R_{p}+h,\alpha \right)\right)}{qR\left(\alpha \right)}\right]}^{2}\sin\left(\alpha \right)d\alpha ,$$
where α is the angle between the axis of the ellipsoid and scattering vector, and $R\left(R_{1},R_{2},\alpha \right)$ is determined by the following equation:(9)$$R\left(R_{1},R_{2},\alpha \right)={\left[R_{1}^{2}sin^{2}\alpha +R_{2}^{2}cos^{2}\alpha \right]}^{1/2},$$
where $R_{e}$ is the equatorial radius and $R_{p}$ is the polar radius. The results of the approximation using the form factor of the ellipsoid are presented in Table 3. The fitted lines practically do not differ from the model fits of the polydisperse core-shell ellipsoids.

In both cases, for the ellipsoid and the polydisperse spherical micelle model, the resulting aggregation number was higher than that obtained in [9].

The analysis of the SANS and SAXS data shows that the core-shell model is applicable in both cases, supporting the assumption of a nearly spherical core-shell structure of Chobimalt micelles in the studied concentration range and solvent conditions. The structural parameters obtained by the least squares fitting of the corresponding models are very similar for both experimental data sets. A comparison of the parameters obtained by the SAXS and SANS for the Chobimalt micelles in H_2_O and in D_2_O shows certain differences in the calculated dimensions of the polar shell of the micelle. Such a difference can be attributed to the different weights of the two terms in the equation of the form factor (Equation (3)) for the SAXS and SANS data fits. The influence of the instrumental smearing, substantially different for the two instruments, although taken into account in a standard way, can also contribute to the observed differences in the two types of experiments.

Comparing the SANS/SAXS data obtained for Chobimalt micelles in pure D_2_O/H_2_O and acidic solutions, it is clearly seen that the acidity of the solvent does not affect the behavior of micelles, because the scattering curves completely overlap for the same detergent concentrations.

It is important to mention that Chobimalt molecules form spherical-like micelles in aqueous solutions, but, for example, an addition of cholesterol to cationic gemini surfactants leads to the formation of bilayer vesicles and the enhanced rigidity of the local environment of the vesicles with an increasing proportion of cholesterol in the mixture, as it was concluded by S. Mondal and co-authors [36].

It should be also noted that the obtained CMC value for the Chobimalt molecules ~3 μM is lower than the CMC for most of the sugar-based surfactants with unsaturated chains, with a CMC of about 0.5 mM, and it differs strongly from other sugar-based surfactants with different stereochemistry of the polar head, for which the CMC varies between 3 and 70 mM [37]. At the same time, the models of polydisperse spherical or slightly ellipsoidal micelles, obtained in the analysis of the SANS and SAXS data, are in good agreement with the literature data for the detergent micelles when ellipsoidal shapes are stated for most compounds [38].

Comparing the behavior of Chobimalt to some other sugar-based surfactant solutions, we can see that the relatively short hydrophilic and hydrophobic parts of Chobimalt lead to compact, spherical, or ellipsoidal micelles, while in the sugar head–alkyl tail surfactants, such as the *n*-tetradecyl-β-d-maltoside (β-C_14_G_2_) studied by Ericsson et al. [39], long cylindrical core-shell micelles form in dilute solutions, followed by the formation of lyotropic crystalline phases at high surfactant contents. Surfactants with shorter alkyl tails, such as octyl-β-maltopyranoside, assemble in spherical micelles, as proven by SANS and SAXS [35].

The present results are interesting in connection with the recently found dose-dependent effect of Chobimalt on insulin amyloid aggregation [10]. Thus, in mixtures with protein, increasing the Chobimalt concentration led to a decrease in the small-angle scattering signal from the amyloid aggregates, indicating the partial destruction of the amyloid fibrils, with a simultaneous appearance of Chobimalt micellar aggregates. In general, two important parameters are responsible for the effect of Chobimalt on protein amyloid fibrils, namely the concentration of additives and also their size/shape, as it was recently shown for fullerene-amyloid mixtures [40]. In the case of Chobimalt, reported here, there is no significant change in the micelle size in the studied concentration range, relevant for protein solutions. This means that the increase in the number of Chobimalt micelles alone can cause the amyloid fibrils disintegration. Quite recently, it was shown by SAXS that proteins can adsorb to the surface of micelles of sugar-based surfactants, forming complexes [25]. Accordingly, the binding of proteins to micelles may be the main mechanism responsible for the decrease in the amount of amyloid fibrils with the increasing number of Chobimalt micelles. Nevertheless, the detailed mechanism of the Chobimalt action on protein requires further structural studies.

## 3. Materials and Methods

### 3.1. Determination of Chobimalt Critical Micelle Concentration Using Fluorescence Probe

The CMC of Chobimalt (Anagrade, CH220, purity > 99%; M_w_ = 1035 Da, Anatrace, OH, USA) was measured using the fluorescence intensity of a fluorescence probe 8-Anilino-1-naphthalenesulfonic acid (ANS, A1028; Sigma Aldrich, St. Louis, MO, USA). Chobimalt samples (150 µL) in concentrations from 0.05 µM up to 100 µM were mixed with 5 µL ANS (from a freshly prepared 5 mM stock solution, 5 mM in 50 mM PBS). Three different experimental approaches were applied and Chobimalt was dissolved in: (i) milli-Q water (denoted as H_2_O); (ii) acidic condition, pH 2.0 (adjusted using HCl, denoted as H_2_O-HCl); (iii) the presence of 100 mM NaCl at pH 2.0 (H_2_O-NaCl). The fluorescence intensity was measured using a 96-well plate by a Synergy MX (BioTek) spectrofluorometer. The excitation was set at 380 nm and the emission was recorded at 485 nm. The excitation and emission slits were adjusted to 9.0/9.0 nm and the top probe vertical offset was 6 nm. Each experiment was performed in triplicates; the error bars represent the average deviation for repeated measurements of three separate samples.

### 3.2. Small-Angle Neutron Scattering (SANS)

SANS measurements were performed at the *Yellow Submarine* diffractometer operating at the Budapest Neutron Center, Hungary [41]. Samples were placed in 2 mm thick Hellma quartz cells. The temperature was set at 20 °C and controlled within 0.1 °C using a Julabo FP50 water circulation thermostat. The range of momentum transfer q was set to 0.07 ÷ 3.1 nm^−1^. The q value is defined as q = 4 π/λ sinθ, where 2θ is the scattering angle. Two configurations with sample-detector distances of 1.15 m and 5.125 m and mean neutron wavelengths of 0.6 nm and 1.04 nm were used to have access to the whole range of q. The raw data were corrected for sample transmission, scattering from the empty cell, and room background. Correction of the detector efficiency and conversion of the measured scattering to an absolute scale was performed by normalization to scattering from water.

### 3.3. Small-Angle X-ray Scattering (SAXS)

SAXS measurements were carried out using an SAXSpoint 2.0 instrument (Anton Paar, Austria). Using Cu Kα radiation and a hybrid photon-counting 2D EIGER R series detector, a q-range of 0.07–5 nm^−1^ was covered with q-resolution δq < 0.003 nm^−1^. The measurements were carried out on samples in solution at 20 °C using a special quartz capillary of 1 mm in diameter.

### 3.4. Sample Preparation

For SANS/SAXS measurements, Chobimalt samples were prepared by dissolving in either D_2_O, D_2_O-DCl, pD~2.0 (SANS) or in H_2_O, H_2_O-HCl, pH~2.0 (SAXS).

## 4. Conclusions

The micelle structure was studied by small-angle neutron and X-ray scattering. The models of the cholesterol core–maltose shell interacting micelles were applied for both the SAXS and SANS data analyses. Polydisperse spherical or moderately elongated ellipsoidal shapes of the Chobimalt micelles were concluded in the studied concentration range, and the micelle aggregation numbers were determined from the calculated micelle volumes. The aggregation numbers in the range of 189–283 were found in the surfactant concentration range of 0.10–6.90 mM. The presence of HCl in the solvent did not exhibit a visible impact on the structure of the Chobimalt micelles. The CMC values of the synthetic surfactant Chobimalt in water and in the acidic aqueous solution were determined by fluorescence measurements and found to be in the range of 2.5–3.0 μM. We can conclude that the present results may serve as reference information in studies of Chobimalt–protein complexes and various sugar-based surfactant solutions [35,36,37,38,39,42,43,44,45,46].

## Figures and Tables

**Figure 1 molecules-28-01811-f001:**
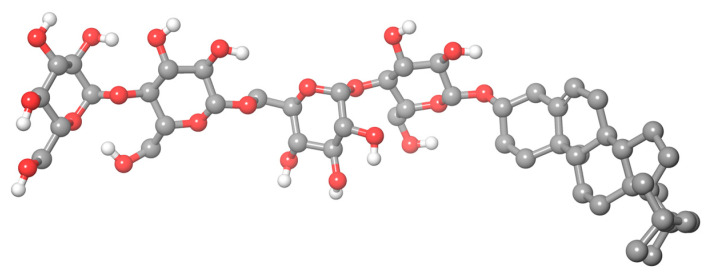
Three-dimensional chemical structure of Chobimalt molecule.

**Figure 2 molecules-28-01811-f002:**
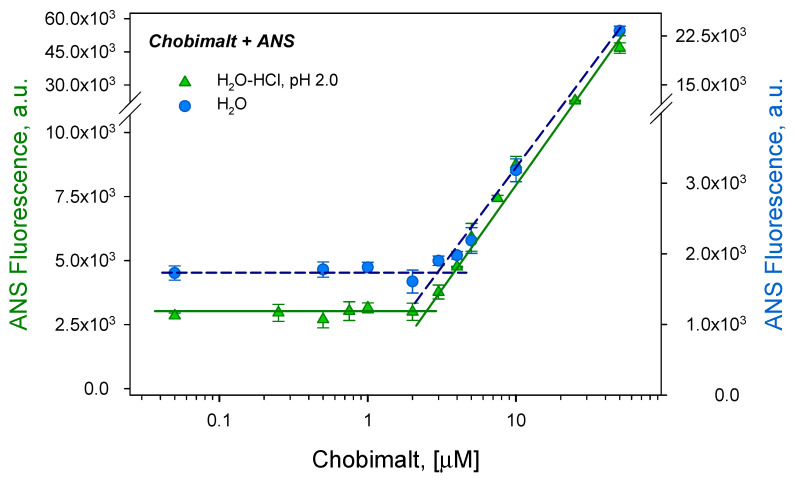
Determination of the critical micellar concentration of Chobimalt using ANS fluorescence assay dissolved in H_2_O (blue circles) and in H_2_O-HCl (green triangles).

**Figure 3 molecules-28-01811-f003:**
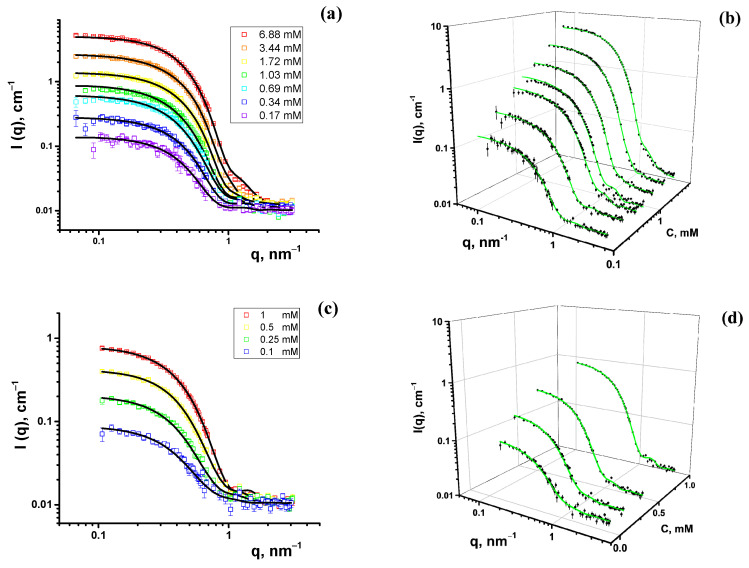
Experimental SANS curves of Chobimalt micelles in D_2_O-DCl in planar (**a**) and 3D (**b**) representations; Chobimalt micelles in D_2_O in planar (**c**) and 3D (**d**) representations. Solid lines represent the model fits of the spherical core-shell form factor.

**Figure 4 molecules-28-01811-f004:**
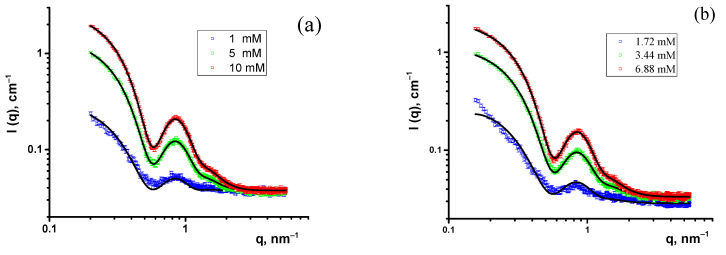
SAXS curves of Chobimalt micelles: (**a**) in H_2_O and (**b**) in H_2_O-HCl.

**Figure 5 molecules-28-01811-f005:**
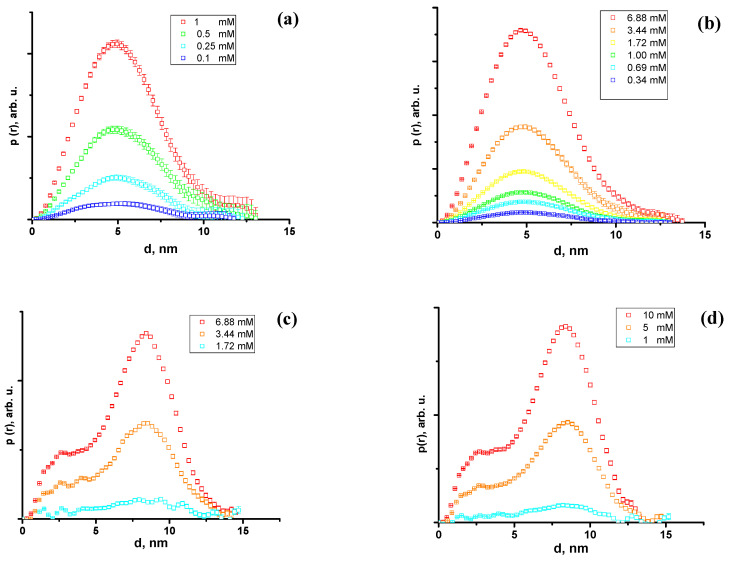
Pair distance distribution functions of Chobimalt micelles in D_2_O (**a**) and D_2_O–DCl (**b**) according to SANS data presented in Figure 3. PDD functions for Chobimalt micelles in H_2_O (**c**) and H_2_O–HCl (**d**) according to SAXS data presented on Figure 4.

**Table 1 molecules-28-01811-t001:** Invariants obtained from IFT analysis of scattering data.

	Sample	C, mM	D_max_, nm	R_g_, nm	I_0_, cm^−1^
SANS	Chobimalt in D_2_O-DCl	0.34	13.3	4.12	0.27
		0.69	13.1	4.15	0.56
		1	13.3	4.07	0.80
		1.72	13.0	4.07	1.35
		3.44	13.2	4.08	2.56
		6.88	14.0	4.06	5.11
	Chobimalt in D_2_O	0.25	12.0	4.21	0.0754
		0.5	12.5	4.17	0.185
		0.75	13.0	4.24	0.416
		1	13.3	4.19	0.779
SAXS	Chobimalt in H_2_O-HCl	1.72	15.0	5.96	0.249
		3.44	14.5	5.58	1.14
		6.88	14.8	5.55	2.13
	Chobimalt in H_2_O	1	14.5	5.25	0.229
		5	15.0	5.57	1.5
		10	13.8	5.45	2.85

**Table 2 molecules-28-01811-t002:** Structural parameters of Chobimalt micelles according to fitting Equation (6) to SANS data using polydisperse spherical core-shell form-factor (Equation (3)). The SLD of the core is fixed (0.22 × 10^10^ cm^−2^).

Sample	C, mM	φ, Vol.%	Core Radius, nm	Shell Thickness, nm	SLD Shell, 10^10^ cm^−2^	PD Index	$N_{agg}$
Chobimalt in D_2_O-DCl	0.17	0.014	1.0(5)	3.1(5)	2.1(3)	0.19(1)	218(33)
	0.34	0.027	1.1(4)	3.1(5)	2.1(2)	0.22(1)	235(32)
	0.69	0.055	1.4(5)	2.9(5)	1.9(2)	0.24(1)	252(32)
	1.00	0.079	1.6(3)	2.8(3)	1.9(1)	0.24(1)	270(18)
	1.72	0.136	1.7(3)	2.7(3)	1.92(5)	0.24(1)	270(18)
	3.44	0.273	1.8(1)	2.6(1)	2.15(5)	0.27(1)	270(7)
	6.88	0.546	1.75(7)	2.72(7)	2.08(2)	0.26(1)	283(4)
Chobimalt in D_2_O	0.10	0.008	1.1(5)	2.8(5)	1.8(5)	0.3(1)	189(35)
	0.25	0.020	1.1(3)	3.1(5)	2.1(3)	0.24(2)	235(27)
	0.50	0.040	1.2(2)	3.1(5)	2.1 (2)	0.25(1)	252(23)
	1.00	0.080	1.5(2)	2.8(5)	2.1(1)	0.25(1)	252(22)

Note: errors are in brackets.

**Table 3 molecules-28-01811-t003:** Structural parameters of Chobimalt micelles according to fitting Equation (6) to SANS data using ellipsoid form factor (Equation (8)). The SLD of the core is fixed (0.22 × 10^10^ cm^−2^).

Sample	C, mM	φ, Vol.%	*R_e_*, nm	*R_p_*, nm	Shell Thickness, nm	Micelle Shell SLD, 10^10^ cm^−2^	$N_{agg}$
Chobimalt in D_2_O-DCl	0.17	0.014	0.9(2)	3.0(2)	3.1(2)	2.3(2)	308(14)
	0.34	0.027	0.9(2)	3.0(2)	3.2(2)	2.1(1)	324(13)
	0.69	0.055	0.9(1)	3.3(1)	3.2(2)	2.1(1)	340(13)
	1.00	0.079	0.9(1)	3.6(1)	3.2(1)	1.92(5)	363(7)
	1.72	0.136	0.9(1)	3.8(1)	3.2(1)	1.95(3)	373(6)
	3.44	0.273	0.9(1)	4.0(1)	3.1(1)	2.10(3)	360(6)
	6.88	0.546	0.92(7)	4.05(3)	3.18(6)	2.06(1)	384(5)
Chobimalt in D_2_O	0.10	0.008	0.8(3)	3.0(3)	3.1(2)	2.2(2)	294(20)
	0.25	0.020	0.9(2)	3.3(3)	3.2(2)	2.3(1)	346(16)
	0.50	0.040	0.9(1)	3.7(1)	3.1(1)	2.3(1)	345(7)
	1.00	0.079	0.9(1)	3.8(1)	3.2(1)	2.32(5)	373(7)

Note: errors are in brackets.

**Table 4 molecules-28-01811-t004:** Structural parameters of Chobimalt micelles according to fitting Equation (6) to SAXS data using polydisperse spherical core-shell form factor (Equation (3)). The SLD of the core is fixed (10.0 × 10^10^ cm^−2^).

Sample	C, mM	φ, Vol.%	Core Radius, nm	Shell Thickness, nm	SLD Shell, 10^10^ cm^−2^	PD Index
Chobimalt in H_2_O	1.0	0.079	4.14(5)	2.13(7)	12.78(8)	0.21(1)
	5.0	0.397	4.48(1)	1.48(2)	14.00(5)	0.17(1)
	10.0	0.793	4.45(1)	1.49(1)	13.97(2)	0.18(1)
Chobimalt in H_2_O-HCl	1.72	0.136	4.42(2)	1.88(5)	13.64(5)	0.22(1)
	3.44	0.273	4.46(1)	1.71(1)	13.33(5)	0.17(1)
	6.88	0.546	4.45(1)	1.61(1)	13.88(1)	0.16(1)

Note: errors are in brackets.

## Data Availability

Experimental data are available from the authors upon reasonable request.

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
