# Peer review of "Micelle Formation in Aqueous Solutions of the Cholesterol-Based Detergent Chobimalt Studied by Small-Angle Scattering"

_molecules, 2023, doi:10.3390/molecules28041811_

Round 1

Reviewer 1 Report

The article by Artykulnyi and coworkers presents a study of a surfactant based on cholesterol.  The research is designed and presented well.  The following matters should be addressed by the authors.

Chobimalt does not seem to be a commonly used surfactant for working with membrane proteins, although the authors have recently studied its interaction with insulin amyloid aggregates.  The authors need to do a better job of motivating why they performed this study.  As it is, it is presented as a characterization of micelles that they used in the previously published work.  This reviewer wonders why the protein-free micelle characterization was not included in the 2022 publication in Frontiers in Molecular Biosciences by the group. 

The sentence in the abstract “Water-soluble cholesterol-based molecule, Chobimalt was developed and synthesized previously by Stanley Howell with colleagues to simulate insoluble cholesterol and, therefore, to facilitate experimental modeling of the biological activity of cholesterol.” should have a reference noted in the format required by the journal of citations in the abstract. 

The authors need to address why the fit of the 1 mM Chobimalt SAXS data presented in Figure 4 is relatively poor compared to the other data sets in the figure. 

Author Response

Chobimalt does not seem to be a commonly used surfactant for working with membrane proteins, although the authors have recently studied its interaction with insulin amyloid aggregates.  The authors need to do a better job of motivating why they performed this study.  As it is, it is presented as a characterization of micelles that they used in the previously published work.  This reviewer wonders why the protein-free micelle characterization was not included in the 2022 publication in Frontiers in Molecular Biosciences by the group. 

Introduction was modified and information about the importance and applications of Chobimalt in the field of membrane protein research has been added. All main changes are highlighted by Yellow in the revised manuscript.

It should be mentioned, that our recent article (2022) about the impact of Chobimalt on protein amyloid fibrils was one of the reasons to initiate a detailed investigation of Chobimalt micellar solutions, which is the topic of the current manuscript. The Chobimalt – amyloid paper focussed on the role of Chobimalt molecules and micelles on the destruction of the protein amyloid fibrils, while the present paper intends to provide a reasonably complete characterization of the micellar solutions of this new type of surfactant.

The sentence in the abstract “Water-soluble cholesterol-based molecule, Chobimalt was developed and synthesized previously by Stanley Howell with colleagues to simulate insoluble cholesterol and, therefore, to facilitate experimental modeling of the biological activity of cholesterol.” should have a reference noted in the format required by the journal of citations in the abstract.

Abstract has been totally rewritten and now there is no literature citation in the revised Abstract.

The authors need to address why the fit of the 1 mM Chobimalt SAXS data presented in Figure 4 is relatively poor compared to the other data sets in the figure. 

The main reason of the visible discrepancy is the very low surfactant concentration, and the magnification of the error obtained in subtraction of the scattering of the water-filled capillary. Fortunately, the deviation is strong only at small angles, where the scattering of the quartz glass is strong. In the medium q range, from which most of the structural information is obtained, the background subtraction is more accurate. For the main results of our study, the higher concentrations of detergent are more important and representative.

Reviewer 2 Report

This current manuscript reports CMC and uses small angle scattering techniques to measure size and shape parameters for Chobimalt, a cholesterol-based water soluble detergent. The original paper that introduced Chobimalt also reported CMC (with the same technique presented here), aggregation number, aggregate size, and inferred spherical shape. The authors have used alternate techniques compared to the original paper, however this study does not add much new information that has not already been reported. The decision to characterize Chobimalt within HCl solutions seems to be of interest to the authors, but I do not see how this is useful to others in the field. In addition, the authors have not discussed their results in the context of the broader field or compared properties of Chobimalt to other micelle systems. For these reasons, I do not find that this manuscript presents significantly novel results or conclusions and is inappropriate for publication in Molecules in its current form.

More specific major concerns are listed below:

-In the introduction, the authors state: “It is intuitive that Chobimalt efficiency is strongly affected by the presence of free detergent molecules whose concentration is restricted by the critical micelle concentration.” I am not able to understand what the authors are trying to communicate here. Please rephrase so that your idea is easily conveyed to the reader. Particularly, please define ‘efficiency’ here, it is unclear what the authors mean.

-What is the motivation for reporting the effect of HCl solutions on Chobimalt micelles? It is clear that the authors used these conditions for their previous studies of amyloids, but how does this interest the broader field? The authors need to state a clear motivation in the introduction. What will this information reveal about the system and how will this be useful for others who wish to use Chobimalt?

-The ANS fluorescence determination of critical micelle concentration is not performed on HCl solutions of Chobimalt, despite the SAXS and SANS experiments being performed with these. For the best comparison, the authors should include CMC determination in HCl solution to demonstrate whether the CMC is altered and for best comparisons among experiments.

- “acid-buffered” is used throughout the manuscript to describe the HCl solution, but to be precise, HCl does not behave as a buffer since it is a strong acid.

-The discussion needs to be greatly expanded. Currently, the discussion is little more than a summary of the results and does not contextualize the finding of the study. Please refer to the Molecules guide for authors for expectations regarding discussion sections:

Authors should discuss the results and how they can be interpreted in perspective of previous studies and of the working hypotheses. The findings and their implications should be discussed in the broadest context possible and limitations of the work highlighted. Future research directions may also be mentioned. This section may be combined with Results.

There is little interpretation in the perspective of previous studies. This manuscript would be greatly improved by providing some context for the results that are presented. As written, it is difficult to understand how these results will help with the use of Chobimalt as in vitro membrane models. As the authors state: “Knowledge of the detergent properties and, in particular, knowledge of the size and structure of micelles in solutions is necessary for the right choice of detergent and conditions for their application for proteins.” Since the authors extract CMC, size, shape, aggregation number, and polydispersity from their experiments, they should compare and contrast their results with those from other detergent micelle systems that are commonly used for in vitro protein studies. In fact, the authors already cite a number of papers that measure these properties for other sugar-based surfactant solutions (references 25-30 and others). This should be included in the discussion and would greatly help other researchers to understand if the use of Chobimalt is appropriate for their application.

-Why was spherical model the only shape model used to interpret the data? It is obvious that the scattering profiles approximate spheres, but there is a possibility that ellipsoids or other nearly spherical shapes may fit the data more precisely. The authors report that polydisperse spheres are the best models to fit the data, but often, ellipsoid models give very similar scattering profiles as polydisperse spheres. For completeness of analysis, authors should fit to alternate models that might feasibly describe the data, then report which model gives the best fit statistics.

Author Response

This current manuscript reports CMC and uses small angle scattering techniques to measure size and shape parameters for Chobimalt, a cholesterol-based water soluble detergent. The original paper that introduced Chobimalt also reported CMC (with the same technique presented here), aggregation number, aggregate size, and inferred spherical shape. The authors have used alternate techniques compared to the original paper, however this study does not add much new information that has not already been reported. The decision to characterize Chobimalt within HCl solutions seems to be of interest to the authors, but I do not see how this is useful to others in the field. In addition, the authors have not discussed their results in the context of the broader field or compared properties of Chobimalt to other micelle systems. For these reasons, I do not find that this manuscript presents significantly novel results or conclusions and is inappropriate for publication in Molecules in its current form.

Introduction and motivation of this work have been improved considerably. The possible application of the Chobimalt and importance of the presented results were discussed. It is the first work in the literature devoted to detailed structural investigation of Chobimalt micelles in water and H2O-HCl solutions, employing small-angle scattering neutron and X-ray methods. Comparison of our SANS/SAXS results with our complementary techniques have been added into revised version of the manuscript.

All main changes in the revised manuscript are highlighted by yellow.

More specific major concerns are listed below:

-In the introduction, the authors state: “It is intuitive that Chobimalt efficiency is strongly affected by the presence of free detergent molecules whose concentration is restricted by the critical micelle concentration.” I am not able to understand what the authors are trying to communicate here. Please rephrase so that your idea is easily conveyed to the reader. Particularly, please define ‘efficiency’ here, it is unclear what the authors mean.

This phrase has been removed in the revised version. The introduction has been rewritten and revised text was highlighted by yellow.

-What is the motivation for reporting the effect of HCl solutions on Chobimalt micelles? It is clear that the authors used these conditions for their previous studies of amyloids, but how does this interest the broader field? The authors need to state a clear motivation in the introduction. What will this information reveal about the system and how will this be useful for others who wish to use Chobimalt?

Importance of HCl solutions for Chobimalt has been described.

-The ANS fluorescence determination of critical micelle concentration is not performed on HCl solutions of Chobimalt, despite the SAXS and SANS experiments being performed with these. For the best comparison, the authors should include CMC determination in HCl solution to demonstrate whether the CMC is altered and for best comparisons among experiments.

ANS fluorescence measurements for Chobimalt in HCl aqueous solutions have been performed according to suggestion, and the results added to Fig.2 of the revised manuscript.

- “acid-buffered” is used throughout the manuscript to describe the HCl solution, but to be precise, HCl does not behave as a buffer since it is a strong acid.

It has been corrected.

-The discussion needs to be greatly expanded. Currently, the discussion is little more than a summary of the results and does not contextualize the finding of the study. Please refer to the Molecules guide for authors for expectations regarding discussion sections:

Authors should discuss the results and how they can be interpreted in perspective of previous studies and of the working hypotheses. The findings and their implications should be discussed in the broadest context possible and limitations of the work highlighted. Future research directions may also be mentioned. This section may be combined with Results.

There is little interpretation in the perspective of previous studies. This manuscript would be greatly improved by providing some context for the results that are presented. As written, it is difficult to understand how these results will help with the use of Chobimalt as in vitro membrane models. As the authors state: “Knowledge of the detergent properties and, in particular, knowledge of the size and structure of micelles in solutions is necessary for the right choice of detergent and conditions for their application for proteins.” Since the authors extract CMC, size, shape, aggregation number, and polydispersity from their experiments, they should compare and contrast their results with those from other detergent micelle systems that are commonly used for in vitro protein studies. In fact, the authors already cite a number of papers that measure these properties for other sugar-based surfactant solutions (references 25-30 and others). This should be included in the discussion and would greatly help other researchers to understand if the use of Chobimalt is appropriate for their application.

Results and Discussions part has been considerably expanded, including new analyses and discussions, and comparisons with structural behaviour of other related surfactants.

-Why was spherical model the only shape model used to interpret the data? It is obvious that the scattering profiles approximate spheres, but there is a possibility that ellipsoids or other nearly spherical shapes may fit the data more precisely. The authors report that polydisperse spheres are the best models to fit the data, but often, ellipsoid models give very similar scattering profiles as polydisperse spheres. For completeness of analysis, authors should fit to alternate models that might feasibly describe the data, then report which model gives the best fit statistics.

Following the recommendation, in the revised manuscript, polydisperse spherical and ellipsoidal core-shell models were used to fit the SANS and SAXS data and the results added into the manuscript. Furthermore, inverse Fourier transformation analysis has been done in order to obtain model-free information on the micelle shape and size. Discussion about used models and results has been added into the revised manuscript.

Reviewer 3 Report

In this work, Artykulnyi et al. examine micelles of a novel cholesterol-based surfactant named chobimalt. The micelles formed in pure water and buffer solutions are examined using SANS and SAXS, after locating the CMC of the surfactant using the fluorescence intensity enhancement of the dye ANS.

The investigation is reasonable, in the sense that this surfactant may be used in formulations used to probe membrane proteins therefore it is important to have basic information about its properties in solution. The problem is that the authors do not carry the analysis of the scattering curves beyond the most basic level. They assume that a core-shell sphere is appropriate for data modeling and they use available software to carry out fits with some assumptions to obtain SLDs. The geometric parameters of the micelles obtained from the SANS and SAXS investigations are different. This the authors choose to ignore, with some hand-waving arguments. In my opinion, the argument below is unacceptable in this type of paper, the only goal of which is to provide structural information on the micelles using scattering methods:

Comparison of the parameters obtained by SAXS and SANS for the acid-buffered H2O and D2O solutions shows certain differences … which can be attributed to the different combination of the SLD levels in case of neutrons and X-rays, which lead to different levels of enhancement or suppression of contribution of the various terms in the form factor (Equation 4) to the reduced chi2. The influence of the instrumental smearing, substantially different for the two instruments, although taken into account in a standard way, can also lead to different results for the two types of experiments.”

As far as I am concerned, this is sufficient to send the paper back to the authors and ask them to do a really thorough analysis of the scattering curves. There are several elements that have not been considered:
(a) Why would the polydispersity be fixed arbitrarily to 20% for all surfactant concentrations? It makes sense to have a variable polydispersity.

(b) The core-shell spherical model should be verified by calculating some of the invariants and PDDF functions at the concentrations used.

(c) The discrepancies of the geometries between SANS and SAXS imply that several different issues should be checked. First, the hydration of the shell must be very significant, and this will affect the SLDs. Since the surfactant is not deuterated the insertion of D2O into the shell would considerably decrease the contrast. Second, the authors should at least check some ellipsoidal models. Finally, the authors should examine what happens if the hard-sphere S(q) is replaced by a sticky hard-sphere model, which is very plausible for surfactants with sugar heads.

Other issues:

-          The abstract and the introduction contain identical paragraphs. This should be corrected.

-          Equation 7 is found at the end of the paper, yet it is the first to appear in the text.

-          What is the φ reported in tables 1 and 2. Clearly it is not the surfactant volume fraction, but it is not explained anywhere.

-          Why are the SLDs changing with concentration? This and the previous equations sound elementary, but I ask them because the authors do not describe clearly the procedures.

Author Response

The investigation is reasonable, in the sense that this surfactant may be used in formulations used to probe membrane proteins therefore it is important to have basic information about its properties in solution. The problem is that the authors do not carry the analysis of the scattering curves beyond the most basic level. They assume that a core-shell sphere is appropriate for data modeling and they use available software to carry out fits with some assumptions to obtain SLDs. The geometric parameters of the micelles obtained from the SANS and SAXS investigations are different. This the authors choose to ignore, with some hand-waving arguments. In my opinion, the argument below is unacceptable in this type of paper, the only goal of which is to provide structural information on the micelles using scattering methods:

Comparison of the parameters obtained by SAXS and SANS for the acid-buffered H2O and D2O solutions shows certain differences … which can be attributed to the different combination of the SLD levels in case of neutrons and X-rays, which lead to different levels of enhancement or suppression of contribution of the various terms in the form factor (Equation 4) to the reduced chi2. The influence of the instrumental smearing, substantially different for the two instruments, although taken into account in a standard way, can also lead to different results for the two types of experiments.”

As far as I am concerned, this is sufficient to send the paper back to the authors and ask them to do a really thorough analysis of the scattering curves.

In the revised manuscript, the analysis of the experimental data has been extended. Different micelle models were used, polydisperse core-shell sphere, and monodisperse core-shell ellipsoids. Further, inverse Fourier transformation analysis has been applied to the data aiming to obtain unbiased information on the scattering particles.

Reflecting to the remark of Reviewer concerning the observed differences in the SANS and SAXS derived structural parameters, we can safely say that the SANS and SAXS data has a significant difference in their “useful” information content, in the sense that the model fitting relies strongly: (i) on the “correctness” of the model, (ii) on the statistical accuracy of the data, (iii) on the contrasts between the different parts of a complex (two-component) particle suspended in a medium, (iv) on the q-resolution of the given instrument and (v) on the accessible q-range. Apart of the last factor, which is nearly the same for the two experiments, the other factors (ii - iv) can lead to a more or less uncontrollable difference in the resulting structural parameters, even if fitting with the same structural model. For these reasons, we think that the moderate differences observed in the two cases are not surprising and can be considered as the overall uncertainty of the given experimental approach, that is extracting structural information from non-ideal experimental data and using idealized particle models. In the revised version we tried to express more clearly this point.

All main changes in the revised manuscript are highlighted by yellow.

There are several elements that have not been considered:
(a) Why would the polydispersity be fixed arbitrarily to 20% for all surfactant concentrations? It makes sense to have a variable polydispersity.

We followed the suggestion of the Reviewer, and have applied variable polydispersities. These parameters are not very accurate and vary in a range of 10-25 % from case to case, which is most likely the result of an overfitting and cannot be considered as exact results for each concentration. Nevertheless, the obtained values lie in a reasonable range. We do not attribute particular importance of the individual polydispersity values obtained by fitting. The results of the new fits with variable polydispersity are included now and all information was updated.

(b) The core-shell spherical model should be verified by calculating some of the invariants and PDDF functions at the concentrations used.

In the revised manuscript, we present results of the IFT calculations before the more detailed model analysis. The PDDF functions resulting from IFT fits are added as new Fig.5 and Table1.

(c) The discrepancies of the geometries between SANS and SAXS imply that several different issues should be checked. First, the hydration of the shell must be very significant, and this will affect the SLDs. Since the surfactant is not deuterated the insertion of D2O into the shell would considerably decrease the contrast. Second, the authors should at least check some ellipsoidal models. Finally, the authors should examine what happens if the hard-sphere S(q) is replaced by a sticky hard-sphere model, which is very plausible for surfactants with sugar heads.

In the new analysis, the hydration of the surfactant shell was considered. Also results of the fits by ellipsoidal model were presented in addition with the previously used spherical model. As the quality of the fit is rather good and the trends of the obtained parameters seems to be reasonable, with increasing of the detergent concentration, we decided not to include one more parameters into the analysis, and avoid using more complicated equation for fitting with a sticky hard-sphere model.

Other issues:

-          The abstract and the introduction contain identical paragraphs. This should be corrected.

The abstract has been totally rewritten and Introduction was substantially modified.

-          Equation 7 is found at the end of the paper, yet it is the first to appear in the text.

The original placements were caused by the required style format of the journal, in which the methodology is place in the end of the article. Now we have rearranged the text and moved equations into the beginning of the Results and Discussions part. This way it looks more natural that firstly we describe equations and then results of the fitting procedure.

-          What is the φ reported in tables 1 and 2. Clearly it is not the surfactant volume fraction, but it is not explained anywhere.

φ parameter is the volume fraction calculated from the actually used weight fraction.

-          Why are the SLDs changing with concentration? This and the previous equations sound elementary, but I ask them because the authors do not describe clearly the procedures.

In the former version, SLD changes could be caused by the influence of the hydration water in the shell. Also, the instability of the fitting procedure could be related to the different statistical accuracy of the data for the different concentrations. In the revised version we tried to clearly explain the model and the parameters.

Round 2

Reviewer 2 Report

This draft of the manuscript is a significant improvement. The authors would improve the quality and reach of the manuscript with further comparisons with other membrane model micelle systems. However, the current form of the manuscript is likely publishable.

Reviewer 3 Report

The authors have carried out sufficient additional analysis to achieve a more complete picture of the structure of Chobimalt micelles. There are still some things that could be examined (e.g. the PDDF is different for SAXS and SANS and its form requires interpretation - also there is a question if a triaxial ellipsoidal model has been applied), but the paper is now acceptable in its present form.